# Electrochemical Gene Amplification Signal Detection of Disposable Biochips Using Electrodes

**DOI:** 10.3390/s22228624

**Published:** 2022-11-09

**Authors:** Gyo-Rim Kim, Ji-Soo Hwang, Jong-Dae Kim, Yu-Seop Kim, Chan-Young Park

**Affiliations:** 1School of Software, Hallym University, Chuncheon-si 24252, Korea; 2Bio-IT Research Center, Hallym University, Chuncheon-si 24252, Korea

**Keywords:** impedance spectroscopy, point of care testing (POCT), label-free detection, PCR chip, biochip

## Abstract

Real-time Polymerase Chain Reaction (RT-PCR), a molecular diagnostic technology, is spotlighted as one of the quickest and fastest diagnostic methods for the actual coronavirus (SARS-CoV-2). However, the fluorescent label-based technology of the RT-PCR technique requires expensive equipment and a sample pretreatment process for analysis. Therefore, this paper proposes a biochip based on Electrochemical Impedance Spectroscopy (EIS). In this paper, it was possible to see the change according to the concentration by measuring the impedance with a chip made of two electrodes with different shapes of sample DNA.

## 1. Introduction

Gene analysis using molecular diagnostic technology has the advantage of high accuracy and high sensitivity through steady research and development. Therefore, gene amplification and analysis techniques using molecular diagnosis are used in various fields such as pathogen detection and disease diagnosis [1]. The epidemic of new infectious diseases such as COVID-19 and monkeypox raises the question of the importance of prompt and accurate diagnosis of infectious diseases along with many social issues in the international community. In particular, along with developing countries such as Africa and South America, the importance of developing POCT (Point-of-Care Test) devices or diagnostic kits that can be easily accessed by households and individuals is emerging. Simple and rapid diagnosis using POCT devices can not only block the spread of domestic infectious diseases but also prevent the inflow and spread of infectious diseases abroad [2,3].

Among infectious disease detection methods, qPCR (Real-Time Polymerase Chain Reaction) is called a ‘gold standard’ and uses a fluorescent label to quantitatively analyze the change in fluorescence brightness during the amplification process in real-time to confirm the presence of infection. However, in the case of using a fluorescent label-based technology, expensive equipment for fluorescence detection is required, and a sample preparation process for analysis and the skills of skilled personnel are required. Accurate temperature control is an important issue in increasing the accuracy of amplification. Due to the disadvantages of the qPCR method, which requires expensive equipment and skilled personnel, it is very disadvantageous in terms of cost and utilization to be used as a diagnostic device for simple and quick use in the field [4,5].

In this paper, we propose a gene amplification method using loop-mediated isothermal amplification (LAMP) and a label-free detection method instead of the real-time polymerase chain reaction method, which is widely used as a conventional gene amplification method as a method for detecting amplified genes.

LAMP is an isothermal gene amplification technology that has been actively studied in place of the polymerase chain reaction method in recent years. Previously, precise temperature control was an important issue in PCR methods. However, precise temperature control is difficult when performing thermal cycling with a 2–3 step protocol (denaturation: 95 °C, junction: 58–60 °C, elongation: 72 °C). Unlike this polymerase chain reaction method, LAMP is amplified at a constant temperature of about 60~65 °C. There is no need for separate temperature control, and sufficient amplification is possible even in a constant-temperature water bath. In addition, since 4 to 6 primers are used, the sensitivity and specificity are high. Above all, since the amplification process is performed at a single temperature, a fast amplification result can be obtained with a simple process [6]. Due to these advantages, the LAMP method can be advantageously applied to the manufacture of on-site diagnostic devices compared to the existing PCR method that requires precise temperature control [7,8,9,10,11,12].

When amplifying a gene using PCR or LAMP method, there is a fluorescence detection method in which a change in fluorescence brightness is compared and confirmed as a method generally used to check the amplification result in real time. In addition, there is a colorimetric method that allows you to easily check the results with the naked eye using a reagent that changes color depending on the characteristics of the solution. However, using the fluorescence detection method, it is necessary to construct a complex optical structure or use expensive optical equipment, which may increase the volume of the product or the price. In addition, the colorimetric method is disadvantageous in obtaining accurate and reliable analysis results [13,14,15,16].

However, the impedance measurement is used in applications such as real-time measurement of cell culture and can be implemented easily and compactly with a simple structure. Therefore, in this study, we want to check whether gene amplification is confirmed by using the characteristic that there is a difference in the measured impedance value according to the difference in concentration. The impedance measurement method has a high correlation with the concentration of the sample solution and the fragment length of the DNA length. In addition, since it does not require an optical detection unit and can implement a detection device having a small and simple structure, it is advantageous to produce a small device for on-site diagnosis [4,17,18,19,20].

In this paper, the impedance of various concentrations of sample solutions was measured by making biochips using two types of printed circuit boards (PCBs), box tapes, double-sided tapes, and plastic films with different electrode materials. It was found that there was a difference when the average was obtained by measuring several times and compared by concentration.

## 2. Materials and Methods

### 2.1. Driving and Detection System

The proposed drive system for gene amplification is divided into a local system part that controls the temperature for isothermal amplification and a host PC part that processes the user interface. When measuring the amplified signal by the conventional fluorescence detection method, the LED, the light source, was controlled by the local system, and the camera module used for detection was connected to the host [21]. However, if the impedance measurement method is used, it can be implemented using a simple circuit.

Figure 1 shows the proposed biochemical measurement system. The sine wave voltage generated by the function generator of the USB oscilloscope (Analog Discovery 2, Digilent Co., Ltd., Pullman, WA, USA) used for the measurement is converted into a current by the input resistance R of the inverting amplifier and flows to the electrode connected to the feedback. The impedance of the sample solution exposed to the electrode of the chip can be determined from the applied voltage (Ch.A) measured using an oscilloscope, the output (Ch.B) of the amplifier, and the reference resistance R. This oscilloscope is connected to a PC through a USB interface and is controlled by a python program developed using a software development kit provided by the company to sample the input and output voltages of the amplifier [22,23].

The proposed driving system for gene amplification is a system for isothermal amplification. It was designed to maintain the amplification temperature (62~65 °C) using PWM control to maintain the temperature for the amplification of nucleic acids.

In this system, temperature control is another important factor, and it is designed to control the temperature by sensing the current temperature and heating it to the target temperature including the heater and fan, and cooling it. The temperature of the chip was set to increase or decrease at about 10 °C/s, and the processing time was set to control the temperature with an error range of less than 0.5 °C within 50 ms. Since it is difficult to set the resolution with the general timer provided by windows, a high-precision event timer was used to set the resolution to less than 1 ms [24].

In the existing polymerase chain reaction, gene amplification was made through a denaturation process at a high temperature of 92~95 °C, a binding process at 58~60 °C, and an elongation process at 72~73 °C. In this process, precise heating and cooling should be possible [21,25,26]. On the other hand, in the case of isothermal amplification, the configuration for temperature control can be simplified because it is only necessary to maintain a constant temperature of 62 to 65 °C [27].

The drive system used in this paper consists of a temperature control unit for isothermal amplification and a connector that can accommodate a biochip containing reagents such as nucleic acids and primers.

As shown in Figure 2, the detection system for determining whether gene amplification is present is designed to detect amplification through impedance change rather than fluorescence detection or colorimetric detection, which are mainly used in the existing polymerase chain reaction and loop-mediated isothermal amplification methods. Therefore, it is possible to miniaturize the detection system of the conventional PCR device using impedance instead of the optical equipment and darkroom required for the existing measurement.

The detection part used in the experiment was designed to measure the impedance of the signal output during the amplification process by connecting a USB oscilloscope to the drive system.

### 2.2. Biochip

Figure 3 shows the presented biochip consisting of a reaction chamber using tapes and plastic on a Printed Circuit Board (PCB). The chip was composed of a total of 4 layers, and packing tape, double-sided tape, and a plastic cover on the printed circuit board constituted a reaction chamber containing the sample for the amplification reaction.

First, a 0.2 T thick matte black flexible PCB was used at the bottom of the chip. There is a heater for heating the reagents in the reaction chamber on the upper surface of the PCB, and a temperature sensor (NTC thermistor) is attached to the lower surface of the PCB so that amplification can occur through temperature control.

The heater was configured in a pattern format. The thermal spread is made of copper to evenly distribute the generated heat, and the temperature can be adjusted to maintain an isothermal state of 62~65 °C. For impedance measurement, the heater pattern part was divided in half to expose the electrode.

A 50 μm thick transparent packaging tape was attached to the printed circuit board. Since the sample may be damaged if the sample is directly exposed to epoxy, a PCB material, a packing tape made of oriented polypropylene was used to ensure biocompatibility. As shown in Figure 4, the double-sided tape was punched into the designed chamber shape and used as a reaction chamber that can contain reagents for the amplification reaction. A 500 μm-thick polycarbonate plastic film was used as the cover of the reaction chamber to fabricate a chip capable of gene amplification.

At this time, a circular hole with a diameter of 2 mm was drilled in the packaging tape and attached to the PCB to prevent the sample from being damaged by excessive contact with the electrode [3].

As shown in Figure 4, two types of impedance measurement chips were fabricated using tin and gold as electrodes to compare the performance of the electrodes.

For these chips, electrodes are included as part of the heating pattern on the top of the PCB for impedance measurement. Figure 4a is a biochip with a heater part coated with white silk material and exposed with tin-plated electrodes with a diameter of 2 mm. On the other hand, Figure 4b shows a biochip in which all exposed electrodes and heaters are plated with gold. The heater area coated with white silk and gold was coated with various materials considering that it is also used for fluorescence detection.

For measurement, the sample reagent must be in direct contact with the electrode, so make a 2 mm diameter hole in the box tape made of ‘Oriented PolyPropylene’ that corresponds to the bottom of the chamber and attach it to the printed circuit board so that the electrode and the reagent come into direct contact with the sample solution concentration It was prepared similarly to the nano drops used for measurement.

Using the chip and drive system proposed in the subsequent experiment, the impedance value of the sample solution exposed to the electrode during the isothermal amplification process is measured in real-time, and the change is analyzed to check whether the gene is amplified.

In fact, the reaction chamber containing the sample reagent and performing the amplification process was manufactured in the shape of a chamber on double-sided tape, and the upper surface was covered with a plastic film made of polycarbonate material. In addition, the housing was made of polycarbonate to fix the tapes and plastic film attached to the PCB.

### 2.3. Materials and Methods

In this paper, to analyze the change in impedance according to the concentration of the sample, 7 types of sample solutions with different concentrations were used without actual isothermal amplification. As the sample DNA used in the experiment, 500 ng of Escherichia virus Lambda DNA was used, and the sample DNA was diluted to 100 ng/μL, 50 ng/μL, 10 ng/μL, 1 ng/μL, 10^−1^ ng/μL, 10^−2^ ng/μL, and 10^−3^ ng/μL with distilled water to make 7 samples with different concentrations. After, in the drive and detection system shown in Figure 5, the impedance of samples with different concentrations was then measured to confirm the value and analyze the changing trend. First, Digilent Co., Ltd.’s software, WaveForms, was used. Using the waveform generator in this software, it was found that the optimal input resistance of each chip was 39 KΩ and 62 KΩ when the two types of chips had an output voltage of 3 V.

To measure the impedance value of the sample by input resistance, inject 36 μL of the diluted sample into two types of biochips with tin and gold-plated electrodes, plug it into the chip connector, connect to the drive system, and start the measurement using a USB oscilloscope. The measured values were stored after sampling until consistent and stable impedance values were seen. The reliability of the measured values was analyzed by measuring the average of 9 times for each concentration on two types of chips, and calculating the standard deviation and standard error based on the measured values.

## 3. Results

Figure 6 shows the results of measuring samples with different concentrations with two types of chips. The experimental results show the impedance measurement values for each concentration when the input resistance is stable and measured at 39 KΩ and 62 KΩ. As shown in Figure 6a,b, it was possible to see the difference in the measured impedance value depending on the material of the electrode of the chip. When the concentration is more than 10 ng/μL, it can be seen that the impedance value is linear for both biochips made of tin-plated and gold-plated. However, in a sample having a concentration of 1 ng/μL or less, a change in impedance values of different aspects can be seen depending on the material of the electrode. Through this, it was found that depending on the electrode material, not only the difference in the magnitude of the impedance value occurs but also the aspect of the impedance value changes.

When the impedance values of samples of different concentrations are measured several times and the average values are plotted as a scatter graph, you can see the unusual value of the impedance measured in the sample with a concentration of 1 ng/μL in chip #2 (gold-plated electrode chip).

It can be seen that the impedance value decreases exponentially in the samples with high concentrations such as 10, 50, and 100 ng/μL. This phenomenon was able to obtain similar results for all of the set input resistances. The reason for obtaining such a result is that since the chip is connected vertically rather than horizontally to the connector during measurement, it is estimated that the reagent is not uniformly exposed to the electrode, resulting in inaccurate values being measured. Therefore, in subsequent experiments, measuring the chip by laying it horizontally seems necessary.

Overall, in Figure 6, there is a slight difference for each input resistance. However, in the case of Chip #1 (tin-plated electrodes), it shows a linear increase until the concentration of the sample is 1 ng/μL, and then linearly decreases after that. However, in the case of Chip #2 (gold-plated electrodes), it can be seen that the average value of the impedance decreases linearly according to the concentration.

In Figure 6, the indicated values represent the average of the measured values. However, in order to analyze the reliability of the values, the standard deviation and standard error were calculated based on the measured values as shown in Table 1. Since the standard error calculated through the values is small, it is not indicated on the plot.

Theoretically, the higher the concentration of the sample, the faster the chemical reaction and the lower the resistance, and the lower, the slower the reaction and the higher the resistance. The gold electrode chip shows that the impedance decreases linearly except for the value at 1 ng/μL.

The minimum sample concentration capable of distinguishing concentrations in the tin and gold electrodes is 10 ng/μL. As can be seen from Table 1, the standard deviation and standard error of the tin electrode are larger than that of the gold electrode. With some exceptions, this shows that gold electrodes are relatively stable compared to tin electrodes.

The currently proposed chip stands upright to perform amplification and detection of samples. In the case of PCR, in the 3-step or 2-step protocol of 95 °C→60 °C→72 °C or 95 °C→60 °C, air bubbles are generated at high temperatures. The reaction chamber is designed so that the air bubbles were collected at the top [21,26]. However, in the case of a chip used for isothermal amplification, there is no need to consider the problem of air bubble generation. The chip will be laid horizontally in subsequent experiments to reduce variables affecting the accuracy, such as the effect of gravity.

Conventional fluorescence detection systems often require complex optical structures or use expensive fluorescence detection devices. These existing commercial systems are very bulky and disadvantageous in terms of price. The proposed system overcomes these limitations and can miniaturize into a simple portable equipment size. The biochip is a disposable chip based on real-time impedance measurement. The electrode is exposed for impedance measurement, and accurate measurement is important for reliable quantitative analysis. Therefore, various subsequent experiments are required to increase the accuracy, such as considering the material of the electrode and the biocompatibility of the tape used.

## 4. Conclusions

This paper confirmed that the measured impedance value showed a difference according to the concentration using the proposed chip and measurement system instead of the fluorescence indicator analysis method and the colorimetric comparison method, which is used often for quantitative analysis of gene amplification.

In addition, it was confirmed that the pattern of the measured impedance was different according to the material of the electrode exposed to the sample solution. The experiment made it possible to classify samples by concentration by measuring the impedance change instead of the conventional analysis method. However, when the concentration was 1 ng/μL or less, the error rate of the impedance value was measured to be large.

It is presumed that the biochip used in the proposed system was connected vertically rather than horizontally to the connector of the driving system, and the reagent and the electrode did not make perfect contact. In a future experiment, we plan to place the chip horizontally and conduct gene amplification tests on the contact between the electrode and the reagent.

In this paper, instead of the fluorescence detection method used in the existing real-time polymerase chain reaction, the process of measuring the impedance of the sample solution was adopted to minimize the on-site diagnosis equipment and simplify the structure. Using two different types of printed circuit boards (PCBs), tapes, and plastic plates, a biochip for measuring impedance for gene amplification and detection for determining whether to amplify or not was produced. And, by measuring the impedance at the electrode exposed to the reagent, the change by concentration was compared and analyzed.

When the impedance of sample solutions of various concentrations was measured using two types of input resistances and two types of electrodes and the average value was plotted as a scatter graph, the impedance decreased exponentially in the sample with a high concentration of 10, 50, 100 ng/μL. It was confirmed through the experiment that the impedance values according to the DNA concentration showed a difference.

In the case of Chip #1, it was confirmed that the impedance measurement value was stably reduced when the concentration of the sample was 10 ng/μL or more, and in the case of Chip #2, it was confirmed that the measurement value was stably measured when the concentration was 1 ng/μL or more. It can be seen that Chip #2, which is entirely plated with gold, shows better results in impedance measurement than Chip #1, which exposes electrodes coated with tin on a heater pattern made of silk material.

In a future experiment, we will try to determine whether the quantitative analysis is possible by comparing the impedance change in real-time during the actual Loop-mediated isothermal amplification process through additional verification experiments related to the direction of the chip. If real-time quantitative analysis of gene amplification is possible in a simple way using a biochip fabricated of PCBs, tape, and plastic film, it is expected that it will show the possibility of realizing a low-cost, ultra-small POCT device.

## Figures and Tables

**Figure 1 sensors-22-08624-f001:**
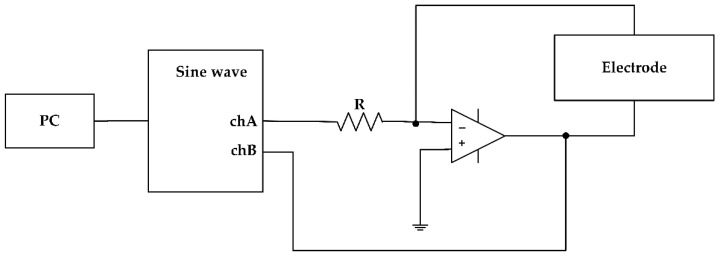
Impedance-based measurement system.

**Figure 2 sensors-22-08624-f002:**
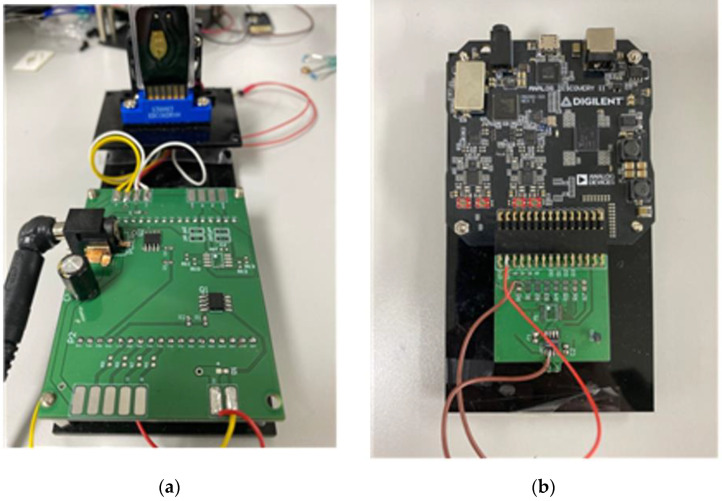
Total driving and detection system (**a**) Driving system (**b**) impedance-based measurement system.

**Figure 3 sensors-22-08624-f003:**
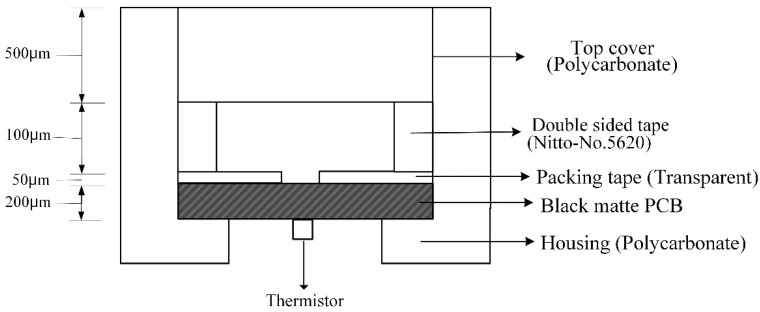
The cross section of the biochip.

**Figure 4 sensors-22-08624-f004:**
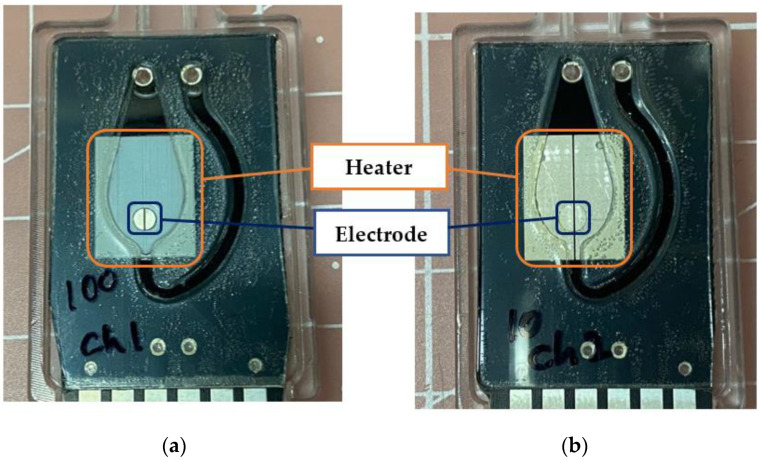
Two kinds of chips for impedance measurement. (**a**) A chip coated with white silk on the exposed electrode with a diameter of 2 mm made of tin and the heater pattern except for the electrode. (**b**) A chip in the form of exposing the electrode by making a hole 2 mm in diameter on the printed circuit board on which both the heater pattern and the electrode are gold-plated.

**Figure 5 sensors-22-08624-f005:**
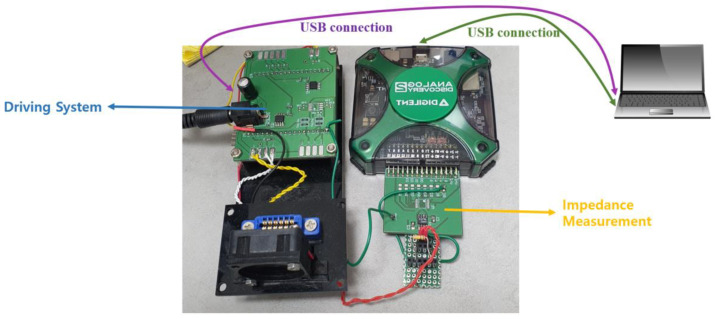
Driving and detection system used in the experiment.

**Figure 6 sensors-22-08624-f006:**
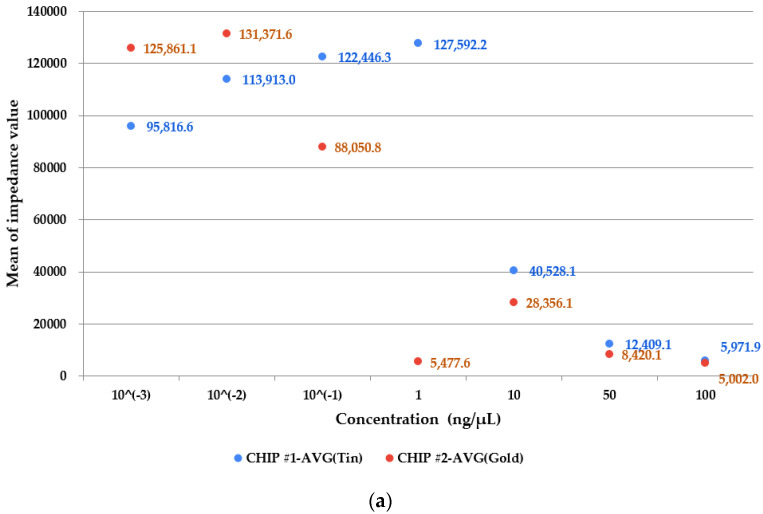
Change of average impedance by sample concentration in two types of chips according to the input resistance. (**a**) Input resistance: 39 Ω (**b**) Input resistance: 62 Ω.

**Table 1 sensors-22-08624-t001:** Analysis according to electrode material.

Chip Number	1 (Tin)						2 (Gold)					
Input Resistance	39 Ω			62 Ω			39 Ω			62 Ω		
Concentration	Mean	Std	Standard Error	Mean	Std	Standard Error	Mean	Std	Standard Error	Mean	Std	Standard Error
10^−3^	95,816.6	72.678	24.226	81269.3	80.671	26.890	125,861.1	14.998	4.999	141,274.5	18.096	6.032
10^−2^	113,913.0	46.615	15.538	77,125.9	160.235	53.412	131,371.6	24.377	8.126	122,118.8	408.235	136.078
10^−1^	122,446.3	47.180	15.727	116,753.2	45.156	15.052	88,050.8	245.800	81.933	85,035.7	33.873	11.291
1	127,592.2	30.725	10.242	147,660.3	93.186	31.062	5477.6	2.639	0.880	4653.0	5.553	1.756
10	40,528.1	33.645	11.215	28,269.7	38.428	12.809	28,356.1	3.593	1.198	27,611.6	23.495	7.832
50	12,409.1	6.543	2.181	9415.3	10.120	3.373	8420.1	44.629	14.876	7098.5	4.350	1.450
100	5971.9	5.829	1.943	5155.5	5.625	1.875	5002.0	2.777	0.926	3973.6	3.066	1.022

## Data Availability

Data sharing not applicable.

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
