# Peer review of "Electrochemical Gene Amplification Signal Detection of Disposable Biochips Using Electrodes"

_sensors, 2022, doi:10.3390/s22228624_

Round 1

Reviewer 1 Report

The idea proposed in the manuscript is quite good with a possible practical use but the whole study appears unfinished based on preliminary data.

- the authors did not properly characterize their biochip: there should be taken the standard analytical procedures include calibration in a proper number of repeats, the limit of detection determination, interference tests, validation to the standard method, etc. 

- the discussion should be added (either as a part of the Results chapter or solitary chapter Discussion) and the relevance of point-of-care biosensors/biochips fully discussed and relevant papers cited. 

I am forced to recommend rejecting the manuscript in its present form.

Reviewer 2 Report

The authors are demonstrating the feasibility of a miniaturized RTPCR system. This is of high interest to the readers but the following points need to be clarified before an understanding of the results and scaling of the phenomena.

The following details need to be added to clarify the results

1) The literature search needs to be expanded to compare this system to other existing ones in the introduction and discussion (i.e https://www.mdpi.com/2227-9040/10/7/269).  There are many emerging candidates in light of the pandemic.

2) Without an independent display of the cyclical temperature profile (existing control) to your proposed systems, the reader does not know whether the source of discrepancies between gold and tin are conductivity-based or random.

3) Are you comparing gold coating and tin coating (Figure 6) or gold coating and silk (Figure 4)?

4) For both plots in Figure 6 there is a net change at 1 ng/ul.  The direction of loading the fluid cannot explain this transition.  For example is there a transition between diffusion and concentration polarization?

5) Have the test been conducted on the same chip (are the microchannels discarded or they have been flushed after each test)? Lack of replication should be discussed at least for future studies. 

Reviewer 3 Report

As the coronavirus that causes the COVID-19 disease spreads across the world, the IAEA, in partnership with the Food and Agriculture Organization of the United Nations (FAO), is offering its support and expertise to help countries use real time reverse transcription–polymerase chain reaction (real time RT–PCR), one of the fastest and most accurate laboratory methods for detecting, tracking and studying the COVID-19 virus.

But what is real time RT–PCR? How does it work? How is it different from PCR? And what does this have to do with nuclear technology? Here’s a handy overview of the technique, how it works and a few refresher details on viruses and genetics.

Real time RT–PCR is a nuclear-derived method for detecting the presence of specific genetic material in any pathogen, including a virus. Originally, the method used radioactive isotope markers to detect targeted genetic materials, but subsequent refining has led to the replacement of isotopic labelling with special markers, most frequently fluorescent dyes. This technique allows scientists to see the results almost immediately while the process is still ongoing, whereas conventional RT–PCR only provides results at the end of the process.

Real time RT–PCR is one of the most widely used laboratory methods for detecting the COVID-19 virus. While many countries have used real time RT–PCR for diagnosing other diseases, such as the Ebola virus and Zika virus, many need support in adapting this method for the COVID-19 virus, as well as in increasing their national testing capacities.

Therefore, the subject matter of the article is very appropriate. In the introductory part, it was necessary to describe the sensors' virus detection chips in more detail. especially those used in medicine.

Round 2

Reviewer 1 Report

The manuscript was improved significantly. I have no other comments on it. 

Reviewer 2 Report

Dear Authors,

Thank you for making the additional clarifications.

Two minor points remain to be addressed:

1) Can you display a plot of the temperature cycles.  A description is offered but it is not sufficient for baseline comparisons?

2) Given the multivariate nature of the experiments, can you please tabulate the variables tested at each stage as well as the metrics?  The explanation of future studies is still vague without the current experimental matrix.
